# Cannabis Use and Cannabidiol Modulate HIV-Induced Alterations in TREM2 Expression: Implications for Age-Related Neuropathogenesis

**DOI:** 10.3390/v16101509

**Published:** 2024-09-24

**Authors:** Bryant Avalos, Jacqueline R. Kulbe, Mary K. Ford, Anna Elizabeth Laird, Kyle Walter, Michael Mante, Jazmin B. Florio, Ali Boustani, Antoine Chaillon, Johannes C. M. Schlachetzki, Erin E. Sundermann, David J. Volsky, Robert A. Rissman, Ronald J. Ellis, Scott L. Letendre, Jennifer Iudicello, Jerel Adam Fields

**Affiliations:** 1Department of Psychiatry, University of California San Diego, 9500 Gilman Dr., La Jolla, CA 92093, USA; bavalosleyva@health.ucsd.edu (B.A.); jkulbe@health.ucsd.edu (J.R.K.); mkford@health.ucsd.edu (M.K.F.); alaird@health.ucsd.edu (A.E.L.); kcwalter24@gmail.com (K.W.); aboustani@health.ucsd.edu (A.B.); esundermann@health.ucsd.edu (E.E.S.); roellis@health.ucsd.edu (R.J.E.); sletendre@ucsd.edu (S.L.L.); jiudicello@health.ucsd.edu (J.I.); 2Alzheimer’s Therapeutic Research Institute, Keck School of Medicine, University of Southern California, 9880 Mesa Rim Road, San Diego, CA 92121, USA; mm_281@usc.edu (M.M.); jf_385@usc.edu (J.B.F.); r.rissman@usc.edu (R.A.R.); 3Department of Medicine, University of California San Diego, 9500 Gilman Dr., La Jolla, CA 92093, USA; achaillon@health.ucsd.edu; 4Department of Neurosciences, University of California San Diego, 9500 Gilman Dr., La Jolla, CA 92093, USA; jschlachetzki@health.ucsd.edu; 5Division of Infectious Diseases, Department of Medicine, Icahn School of Medicine at Mount Sinai, New York, NY 10029, USA; david.volsky@mssm.edu; 6Department of Neuroscience, Icahn School of Medicine at Mount Sinai, New York, NY 10029, USA

**Keywords:** TREM2, HIV, neuroinflammation, cannabis, immunomodulatory, CBD

## Abstract

Triggering receptor expressed on myeloid cells 2 (TREM2) is involved in neuroinflammation and HIV-associated neurocognitive impairment (NCI). People with HIV (PWH) using cannabis exhibit lower inflammation and neurological disorders. We hypothesized that TREM2 dysfunction mediates HIV neuropathogenesis and can be reversed by cannabinoids. EcoHIV-infected wildtype (WT) and TREM2^R47H^ mutant mice were used to study HIV’s impact on TREM2 and behavior. TREM2 and related gene expressions were examined in monocyte-derived macrophages (MDMs) from PWH (*n* = 42) and people without HIV (PWoH; *n* = 19) with varying cannabis use via RNA sequencing and qPCR. Differences in membrane-bound and soluble TREM2 (sTREM2) were evaluated using immunocytochemistry (ICC) and ELISA. EcoHIV increased immature and C-terminal fragment forms of TREM2 in WT mice but not in TREM2^R47H^ mice, with increased IBA1 protein in TREM2^R47H^ hippocampi, correlating with worse memory test performance. TREM2 mRNA levels increased with age in PWoH but not in PWH. Cannabidiol (CBD) treatment increased TREM2 mRNA alone and with IL1β. RNA-seq showed the upregulation of TREM2-related transcripts in cannabis-using PWH compared to naïve controls. IL1β increased sTREM2 and reduced membrane-bound TREM2, effects partially reversed by CBD. These findings suggest HIV affects TREM2 expression modulated by cannabis and CBD, offering insights for therapeutic strategies.

## 1. Introduction

People with HIV (PWH) continue to experience neurological disorders even with the widespread implementation of suppressive antiretroviral therapy (ART). Despite the therapeutic success of ART in PWH, HIV-associated neuroinflammation can persist and contribute to neurocognitive impairment (NCI). The current estimates of PWH with comorbid HIV-associated NCI range from 20% to greater than 50% in some countries [1,2,3]. The Center for Disease Control and Prevention (CDC) recently estimated that over 53% of PWH in the United States were aged 50 and older [4]. As such, PWH are susceptible to NCI as well as age-related neurological disorders, including deficits in learning, memory, attention, executive function, and motor skills [5,6]. Evidence of premature aging in PWH also includes abnormalities in white matter, increased levels of beta-amyloid, mitochondrial dysfunction, reactive astrocytes and microgliosis [7]. Given the high prevalence of NCI, and the increasing age of the population of PWH, novel in vivo, in vitro and ex vivo models are needed to provide mechanistic insights and to test therapeutic strategies.

Brain macrophages are major contributors to the chronic neuroinflammation in HIV infection and aging [8,9], and they populate the brain as three distinct cell types: microglia, perivascular macrophages, and monocyte-derived macrophages (MDMs) [10]. Microglia and perivascular macrophages are resident in the parenchyma of the brain, whereas MDMs traverse the blood–brain barrier and enter the brain during disease states, including early during HIV infection [10,11]. Models for HIV-induced neurotoxicity, including the novel EcoHIV murine model [12], indicate that proinflammatory brain macrophages are implicated in the neuropathogenesis of HIV [13,14]. Brain macrophages also contribute to homeostasis by modulating the inflammatory signaling and phagocytosing extracellular protein aggregates such as beta-amyloid and dying neurons, processes that may be disrupted in the brains of PWH, especially in those of increased age [15,16,17,18,19]. Brain macrophages produce and respond to cytokines that can contribute to apoptosis and neuronal injury, including the pro-inflammatory cytokine interleukin-1 beta (IL1β) which is associated with neurotoxicity in PWH and in aging-related neurodegenerative diseases [20]. Thus, dampening the pro-inflammatory response of macrophages may provide potential therapeutic avenues to restore tissue homeostasis in the aging or HIV-infected brain.

Triggering receptor expressed on myeloid cells 2 (TREM2) is a transmembrane receptor primarily expressed on macrophages in the brain and in peripheral tissues [21]. Additionally, TREM2 is involved in the recognition and clearance of dying neurons and protein aggregates such as Aβ [21,22,23,24,25,26,27,28,29,30,31,32], both of which are associated with HIV-associated neurocognitive impairment and AD [17,33,34,35,36,37]. TREM2 has multiple isoforms resulting from alternative splicing and proteolytic cleavage, each with distinct molecular weights and functional characteristics. The immature form of TREM2 (~35 kDa), a precursor protein primarily localized in the endoplasmic reticulum, is not fully functional but is essential for proper folding, maturation, and transportation to the cell surface [38]. Once membrane-bound, mature TREM2 (~50 kDa) can bind to ligands, such as lipids and lipoproteins, to initiate intracellular signaling that leads to microglial survival, proliferation, phagocytosis, and anti-inflammatory response functions [39,40,41]. Soluble TREM2 (sTREM2) is generated through the proteolytic cleavage of membrane-bound TREM2 by proteases, such as ADAM10 and ADAM17, and also results in the generation of C-terminal fragment (~10 kDa) [42]. sTREM2 is capable of binding to ligands that would otherwise interact with membrane-bound TREM2, thus modulating inflammatory responses and microglial activity [43]. Elevated sTREM2 has been observed in the cerebrospinal fluid (CSF) and plasma of individuals with various neurodegenerative diseases, including AD and multiple sclerosis [44,45,46]. Moreover, the TREM2 R47H gene variant, a single nucleotide polymorphism that results in an amino acid substitution at position 47 [47], is a loss-of-function mutation that has also been associated with an increased risk of developing AD [48,49]. Individuals with the TREM2 R47H mutation may experience more severe neuroinflammatory responses in the presence of HIV, potentially leading to a higher risk of developing HAND and other neurocognitive impairments [16]. Thus, understanding the function and regulation of TREM2, including in the context of the R47H mutation, is crucial for developing therapeutic strategies for both aging-related neurodegenerative diseases and HIV-associated neurocognitive impairment.

Cannabis use in PWH has previously shown therapeutical potential in managing HIV-associated complications [50]. Recent studies have reported that PWH using cannabis have less immune cell activation compared to PWH that do not use cannabis [51,52]. Cannabis use in PWH is also associated with reduced inflammatory biomarkers in circulation [53], decreased viral DNA found in tissues [54], and a lower prevalence of neurocognitive impairment [55,56]. Delta-9-tetrahydrocannabinol (THC) and cannabidiol (CBD) are cannabinoids found in cannabis that have demonstrated promising anti-inflammatory properties crucial for managing neuroinflammation linked to neurodegenerative diseases [57,58]. CBD, known for lacking the psychoactive properties of THC, is particularly associated with significant anti-inflammatory effects by modulating the immune response to reduce inflammation and protect neural tissue from damage [59,60]. Additionally, activation of the cannabinoid type-2 receptor (CB2) on macrophages may reduce the production of pro-inflammatory cytokines while also promoting microglial motility towards injury sites [61,62]. Targeting cannabinoid receptor 2 (CB2) on peripheral immune cells may reduce the inflammatory mechanisms implicated in HAND, suggesting a pathway for therapeutic intervention [63]. However, the precise pathways that mediate the anti-inflammatory and neuroprotective effect of cannabinoids is not fully understood, particularly in the context of PWH that use cannabis while on suppressive ART.

We hypothesized that cannabinoids like CBD promote TREM2 pathway signaling, and that cannabis use in PWH can reverse the TREM2 dysfunction that mediates HIV neuropathogenesis. To begin investigation on the effect of HIV on TREM2 signaling, we used novel in vivo murine models. This is the first study to combine EcoHIV with the aging-related TREM2 R47H (TREM2^R47H^) mutant mouse model. This study is also the first of its kind to investigate the role of TREM2 in MDMs generated from a cohort of PWH with variable cannabis-use patterns. We examined whether HIV induces alterations in TREM2 and related gene expression and assessed the impact of cannabis use and cannabinoids on TREM2-related changes. The findings presented here reveal that HIV affects TREM2 signaling, and this is modulated by cannabis use, opening up much needed new avenues for therapeutic targeting in PWH on ART.

## 2. Materials and Methods

### 2.1. Ethics

Animal studies were conducted in certified animal research facilities at the University of California, San Diego. These studies also abide by the animal care guidelines set in place by the Institutional Animal Care and Use Committee (IACUC) with full compliance with NIH guidelines. Informed written consent was obtained from all participants of this study in compliance with the Institutional Review Board (IRB) approval at the University of California, San Diego.

### 2.2. Animals and Treatments

To determine if HIV infection induces learning and memory deficits in mice with the TREM2 R47H gene mutation, we utilized 11–12-month-old C57BL/6 wildtype (WT) and mutant TREM2^R47H^ mice that were inoculated with saline (vehicle control) or EcoHIV (2.0 μg p24/mouse) via intraperitoneal (IP) injection. Behavioral testing was performed 4 weeks after treatment of all animals (total of 23 mice; *n* = 5–6 per condition) and brain tissues were collected after completion of behavior tests. Total activity memory (TAM) testing was performed using a high-density Kinder Smart Frame cage rack system (Kinder Scientific, Poway, CA, USA), which continuously monitors movement in X, Y, Z coordinate planes within the chamber using a 7 × 5 beam configuration. On the day of testing, animals are transported in their home cages to the behavioral testing room for acclimation. Duration of test sessions was 10 min and all animals were tested for three consecutive days during the habituation phase, which was followed by a 72 h gap before the final day of testing.

### 2.3. Western Blot Detection of TREM2

Mouse brain tissues were processed for TREM2 immunoblotting as previously described [16]. Briefly, frontal cortex tissues from mouse brains (100 μg) were sonicated in lysis buffer (1.0 mM HEPES, 5.0 mM benzamidine, 2.0 mM β-mercaptoethanol, 3.0 mM EDTA, 0.5 mM magnesium sulfate, 0.05% sodium azide, pH 8.8) containing phosphatase inhibitor (Merck Millipore, Burlington, MA, USA; #524624) and protease inhibitor (Merck Millipore; #04693116001) cocktails. Samples were centrifuged at 2000 rpm at 4 °C for 5 min before collection of supernatants containing whole lysates. Quantification of protein was conducted using a PierceTM Bicinchoninic Acid (BCA) Protein assay kit (Thermo Scientific, Waltham, MA, USA; #23225). Protein lysates in 1X Laemmli Sample Buffer (Bio-Rad, Hercules, CA, USA; #1610747) were vortexed, spun down, and incubated in a hot water bath at 95 °C for 5 min. Samples were then loaded (15 μg protein per well) onto a 4–15% CriterionTM TGX Stain-Free Protein Gel (Bio-Rad; #5678085) and electrophoresed at 200 V for 45 min in Tris/Glycine/SDS running buffer (Bio-Rad; #1610772). Protein gels were transferred using the Trans-Blot Turbo Transfer System that includes PVDF membranes, transfer stacks, and transfer buffer from a Trans-Blot Turbo RTA transfer kit (Bio-Rad; #1704272). Gels were rinsed with water before imaging for total protein using a stain-free blot setting on a Bio-Rad ChemiDoc imager. Membranes were placed in 1X TBS 1% Casein Blocking Buffer (Bio-Rad; #1610782) for one hour at room temperature before overnight incubation at 4 °C with primary antibody for rabbit anti-TREM2 (Thermo Scientific; #PA5-87933; 1:1000). Following removal of primary antibody, blots were washed in 1X PBS for 5 min before adding HRP-conjugated goat anti-rabbit IgG secondary antibody (Bio-Rad; #1706515; 1:5000) for 1 h at room temperature. SuperSignal^®^ West Femto enhanced chemiluminescent substrate (Thermo Scientific; #TG26840A) was applied for visualization of protein bands on membranes. The blots were then stripped using Western Blot Stripping Buffer (Thermo Scientific; #21059) and re-probed with a mouse monoclonal antibody against β-actin (ACTB; Sigma-Aldrich, St. Louis, MO, USA; #A5316; 1:2000) as a loading control. Images were captured, and semi-quantitative analysis was performed on Image Lab Software (Bio-Rad v6.1). Average adjusted total band intensities of total TREM2 protein (mature; 50 kD + immature; 35 kD + CTF; 10 kD) were normalized to WT-saline controls.

### 2.4. Immunohistochemistry of IBA1 and GFAP in Mouse Brain Tissue

To determine if HIV infection affects levels of microgliosis and astrogliosis, mouse brains were collected following behavioral analyses and fixed in 4% paraformaldehyde (PFA) for 5 days. Free-floating 40 μm thick vibratome sections of mouse brains were washed with phosphate-buffered saline with Tween 20 (PBST) three times for 5 min each. The sections were then pre-treated with 3% H_2_O_2_ in PBST 1% Triton X-100 for 20 min to block endogenous peroxidase activity. Following pretreatment, sections were blocked with 2.5% horse serum (Vector Laboratories, Newark, CA, USA; #2-2012) for 30 min at room temperature. Primary antibodies for IBA1 (Wako Chemicals, Richmond, VA, USA; #019-19741; 1:1000) and GFAP (Sigma-Aldrich; #G3893; 1:500) were applied to the sections and then incubated overnight at 4 °C. Primary antibodies were removed the next day and sections were washed three times with PBST. Sections were then incubated with the appropriate secondary antibody, ImmPRESS HRP anti-rabbit IgG (Vector Laboratories; #MP-7401) or ImmPRESS HRP anti-mouse IgG (Vector Laboratories; #MP-7402), for 30 min at room temperature on a rocker. Sections were washed, treated with NovaRED Peroxidase Substrate (HRP; Vector Laboratories; #SK-4800), and incubated until the desired stain was achieved. Control sections were incubated with secondary antibodies only. After staining, tissues were mounted on SuperFrost Plus slides (VWR International, Radnor, PA, USA: #48311-703) using diH_2_O and dried in the dark for 1 h. Coverslips were applied with Vectashield mounting media (Vector Laboratories; #H-1000-10). Immunostained sections were imaged on a digital Olympus microscope and immunoreactivity for IBA1 and GFAP was quantified using Image-Pro Plus Software v7.0 (Media Cybernetics, Silver Spring, MD, USA). Average intensity of the immunostaining in areas of interest in the hippocampus (e.g., CA1, CA2/3, DG regions) were corrected for average background levels obtained from control sections processed without primary antibody. Optical density values for both IBA1 and GFAP within each of the regions were then normalized to the respective WT-saline controls for relative quantification.

### 2.5. Study Population

This study recruited people with HIV (PWH) and people without HIV (PWoH) with varying demographic characteristics (e.g., age, sex). All PWH on stable antiretroviral therapy (ART) for at least six months were considered virally suppressed. Participants were grouped based on their HIV status and cannabis-use patterns following recruitment in San Diego, CA, USA. To account for variability in cannabis-use characteristics (e.g., frequency, quantity, mode of administration, cannabinoid content), both laboratory measures and self-report questionnaires were used to comprehensively characterize current and lifetime cannabis use. Prior to the assessment, current cannabis users were asked to adhere to their regular use pattern to mitigate potential withdrawal effects. Cannabis-use groups were categorized as naive (never used or used ≤6 times/year with ≥60 days of abstinence), moderate (one to six uses a week), or daily (seven days a week). In addition to comprehensive medical and neurobehavioral assessments, participants also underwent venous blood collection. Individuals who tested positive for substances other than cannabis were excluded or rescheduled to minimize potentially confounding effects of acute substance use. Additional exclusion criteria include uncontrolled medical, psychiatric, or neurological conditions; a DSM diagnosis of moderate to severe drug use disorder other than cannabis within the past five years, or mild use disorder within the past six months (excluding tobacco); moderate to severe alcohol use disorder within the past 12 months; safety contraindications for MRI; renal insufficiency, which may increase the risk for nephrogenic systemic fibrosis associated with gadolinium-based contrast agents; allergy to gadolinium-based contrast agents; pregnant or breastfeeding. A total of 55 eligible participants (see Table 1) were included in data from this study.

### 2.6. Separation and Treatment of Monocyte-Derived Macrophages

Peripheral blood mononuclear cell (PBMC) isolation was performed on donor blood by Ficoll gradient separation. Briefly, 15 mL of donor blood was slowly layered onto 15 mL of HISTOPAQUE-1077 (Sigma-Aldrich; #10771) and centrifuged at 400× *g* for 30 min. The monocyte layer was collected, washed with 1X PBS, and centrifuged at 250× *g* before resuspending cells in Iscove’s Modified Dulbecco’s Medium (IMDM; Gibco, Waltham, MA, USA; #12440053) supplemented with 10% human serum (Sigma-Aldrich; #H5667), 1% penicillin/streptomycin (Gibco; #15140122). Automated cell counting was performed on a Countess™ 3 FL (Thermo Scientific; #AMQAF2000) using 0.4% trypan blue solution (Thermo Scientific; #T10282). Cells were plated in 24-well plates (Corning Inc., Corning, NY, USA; Costar #3524) at 400,000 cells/well for RNA testing or 96-well plates (Thermo Scientific; #164588) at 100,000 cells/well for protein testing. Cells were maintained and differentiated in a humidified incubator at 5% CO_2_ and 37 °C for 7 days before receiving experimental treatment. Monocyte-derived macrophages (MDMs) were pre-treated for one hour with cannabidiol (CBD; Cerilliant Corporation, Round Rock, TX, USA; #C-045; 30 µM) before incubating with IL1β (Invivogen, San Diego, CA, USA; #6409-44-01; 20 ng/mL) for 6 h prior to RNA isolation or 24 h prior to fixation and immunostaining. Additional pre-treatments for inhibitor studies include AM 251 (CB1 antagonist; Tocris Bioscience, Minneapolis, MN, USA; #1117; 10 µM), AM 630 (CB2 antagonist; Tocris Bioscience; #1120; 10 µM), GW 6471 (PPARα antagonist; Cayman Chemical, Ann Arbor, Michigan, USA; #11697; 10 µM), or GW 9662 (PPARγ antagonist; Cayman Chemical; #70785; 10 µM). Supernatants were collected and MDMs were washed with 1X PBS prior to RNA isolation or fixation with 4% paraformaldehyde (PFA) solution in PBS (Thermo Scientific; #J19943K2).

### 2.7. Real-Time Quantitative Polymerase Chain Reaction (qPCR) and RNA-Sequencing

RNA isolation was performed according to manufacturer’s instructions via Qiagen RNeasy Plus Mini Kit (Qiagen Inc., Valencia, CA, USA; #74136). Total RNA from MDMs was used for RNA-sequencing and RT-qPCR after being analyzed for purity and concentration with a spectrophotometer. RNA was reverse transcribed to cDNA using the High-Capacity cDNA Reverse Transcription Kit (Applied Biosystems, Waltham, MA, USA; #4368814) according to the kit instructions. Multiplex relative quantification assays were performed on a QuantStudio 3 Real-Time PCR machine (Applied Biosystems; #A28567) using TaqMan Fast Advanced Master Mix for qPCR (Applied Biosystems; #4444557) and individual probes for TREM2 (Hs00219132_m1; #4351370), CHIT1 (Hs00185753_m1; #4448892), SMAD3 (Hs00969210_m1; #4453320), ZAP70 (Hs00896345_m1; #4448892), TREM1 (Hs00218624_m1; #4453320), and VSIG4 (Hs00200695_m1; 4453320). For each assay, ACTB (Applied Biosystems; #4310881E) was used as the endogenous control, and the fold change in gene expression was quantified via the comparative Ct method, as previously described [64]. All donor samples were run in technical duplicates. RNA-sequencing was performed at the UC San Diego IGM Genomics Center utilizing an Illumina NovaSeq 6000 that was purchased with funding from a National Institutes of Health SIG grant (#S10 OD026929). The libraries were sequenced on a NovaSeq S4 flow cell, with initial quality control checks, including per-base sequence quality, GC content, and sequence duplication levels conducted using FastQC (v0.11.9). Clean reads were aligned to the hg38 human reference genome using STAR aligner (v2.7.9a) with default parameters on the Illumina BaseSpace Sequence Hub platform. Differential expression analysis was conducted using DESeq2 (v1.30.0) in R (v4.0.5), with genes displaying a false discovery rate (FDR) < 0.05 considered significantly differentially expressed. Gene ontology (GO) and pathway enrichment analyses were performed using PANGEA (v1.1) and identified significantly enriched GO terms and pathways with a corrected *p*-value < 0.05.

### 2.8. Detection and Measurement of sTREM2

Supernatants collected from treated MDMs were processed to determine levels of sTREM2 using a Human TREM-2 ELISA Kit (Sigma-Aldrich, St. Louis, MO, USA; # RAB1091). Samples were assessed in duplicate and absorbance was measured at 450 nm using a Synergy HTX plate reader (BioTek Instruments Inc., Winooski, VT, USA).

### 2.9. Immunocytochemistry for TREM Detection on MDMs

Cells were washed twice with 1X PBS and subsequently fixed with PFA (4% *w*/*v*) for 20 min at 4 °C. PFA was removed and followed by two washes with 1X PBS. The fixed cells were then incubated with blocking buffer (5% BSA, 0.2% Triton-X in 1X PBS) for one hour at room temperature. Primary antibodies for TREM2 (Thermo Scientific; #PA5-87933) were added in blocking buffer (1:250 dilution) and cells were incubated overnight at 4 °C. After primary antibody incubation, cells were washed three times with 1X PBS. Alexa Fluor conjugated secondary antibodies (Invitrogen, Carlsbad, CA, USA; #A21039) in 1X PBS (1:500 dilution) were then added and incubated for 30 min at room temperature on a shaker. Secondary antibodies were removed before cells were washed twice with 1X PBS prior to being incubated with DAPI solution (Thermo Scientific; #D3571; 1:10,000 dilution in 1X PBS) for 5 min. The cells were washed once more then kept in 1X PBS during fluorescent imaging on a CellInsight CX5 High Content Screening (HCS) Platform (Thermo Scientific; #CX51110). Images were captured at 20× magnification on four fields (500 × 500 µM) per well and processed using HCS Studio™ Cell Analysis Software v5.0.

### 2.10. Statistical Analysis

All data are presented as mean ± SEM with statistical analyses that include one-way and two-way ANOVA with Holm–Sidak post hoc multiple comparison tests when appropriate, unless stated otherwise. Statistical significance was determined at *p* < 0.05 for all data, with individual *p*-values reported when near the significance threshold. Experimental sample sizes, as well as normalization of data to specific controls, are stated in each figure legend. Data was analyzed on GraphPad Prism 10.0 software (San Diego, CA, USA).

## 3. Results

### 3.1. EcoHIV Reduces Levels of TREM2 and Alters Memory in Wildtype and TREM2^R47H^ Mice

First, we utilized WT mice and mice with the R47H loss-of-function mutation (TREM2^R47H^) to determine if EcoHIV alters TREM2 protein levels. Immunoblot analyses were performed to determine total expression levels of TREM2 in the frontal cortex brain tissue of WT and TREM2^R47H^ mice treated with EcoHIV or saline (Figure 1A). Densitometry analyses for TREM2 protein normalized to ACTB revealed a reduction in total TREM2 in lysates from EcoHIV-infected WT mice, but no difference in lysates from EcoHIV-treated TREM2^R47H^ mice (Figure 1B). When assessing the ratio of intensity between immature and mature bands from TREM2, EcoHIV-treated WT mice had a significantly higher immature/mature isoform ratio compared to saline controls as well as lysates from TREM2^R47H^ mice (Figure 1C). Similarly, the ratio of intensity between bands for the CTF isoform and total TREM2 was significantly higher in EcoHIV-treated WT mice as compared to saline controls and the CTF:TREM2 lysate ratios from TREM2^R47H^ mice (Figure 1D). To begin assessing the impact of EcoHIV treatment on learning and memory, WT and TREM2^R47H^ mice behavioral testing for three consecutive days of habituation before a final test 72 h after habituation. Total activity memory (TAM) testing assessed exploratory behavior, which showed increased beam breaks in EcoHIV-treated WT and TREM2^R47H^ mice compared to saline-treated WT controls (Figure 1E). Total distance traveled during behavioral testing also showed increased activity in EcoHIV-treated WT and TREM2^R47H^ mice compared to saline-treated WT mice (Figure 1F). Collectively, we show that EcoHIV infection reduces total TREM2 protein levels in the frontal cortex of WT mice, but not in TREM2^R47H^ mice. Furthermore, EcoHIV-treated WT mice exhibit a higher ratio of immature/mature TREM2 isoforms and an increased CTF:TREM2 ratio compared to saline-treated controls and TREM2^R47H^ mice. Behaviorally, both EcoHIV-treated WT and TREM2^R47H^ mice display increased exploratory behavior and greater total distance traveled in the TAM test compared to saline-treated WT controls.

### 3.2. EcoHIV Increases Hippocampal IBA1 Expression in TREM2^R47H^ Mice

To evaluate the status of microglial activation following infection with EcoHIV in WT and TREM2^R47H^ mice, we immunostained vibratome sections of mouse brains with antibody for IBA1 using NovaRed for visualization in the hippocampus (HC). The areas analyzed include the following hippocampal sub-regions: CA1, CA2/3, and DG. In the CA1 region of the HC, the IBA1 signal was less intense in the saline-treated TREM2^R47H^ mice compared to WT mice and EcoHIV-treated TREM2^R47H^ mice (Figure 2A). The quantification of relative optical density in the CA1 region of saline-treated TREM2^R47H^ mice was decreased by ~69% compared to saline-treated WT controls (Figure 2B). Relative optical density for IBA1 in the CA1 hippocampal sub-region of EcoHIV-treated TREM2^R47H^ mice was ~476% greater compared saline-treated TREM2^R47H^ mice (Figure 2B). In the CA2/3 region of the HC, signal intensity for IBA1 was also decreased in saline-treated TREM2^R47H^ mice, as compared to WT and EcoHIV-treated TREM2^R47H^ mice (Figure 2C). Compared to saline-treated WT controls, relative optical density for IBA1 in the CA2/3 hippocampal sub-region decreased by ~70% in saline-treated TREM2^R47H^ mice (Figure 2D). In contrast, EcoHIV treatment increased IBA1 levels in the CA2/3 hippocampal sub-region of TREM2^R47H^ mice by 488%, as compared to saline-treated TREM2^R47H^ mice (Figure 2D). In the DG region of the HC, IBA1 signal was less intense in the saline-treated TREM2^R47H^ mice compared to all other groups (Figure 2E). The quantification of corrected optical density in the DG hippocampal sub-region showed a ~66% reduction in IBA1 in saline-treated TREM2^R47H^ mice compared to saline-treated WT mice (Figure 2F). The relative optical density for IBA1 in the DG hippocampal sub-region of EcoHIV-treated TREM2^R47H^ mice was ~517% greater than saline-treated TREM2^R47H^ mice (Figure 2F). Overall, EcoHIV infection led to differential microglial activation in the hippocampus of WT and TREM2^R47H^ mice. Specifically, EcoHIV-infected TREM2^R47H^ mice exhibited markedly increased IBA1 levels in the CA1, CA2/3, and DG hippocampal sub-regions. These results collectively indicate that heightened microglial activation is associated with the TREM2 R47H gene variant following EcoHIV exposure.

### 3.3. Hippocampal GFAP Expression Is Increased in TREM2^R47H^ Mice Infected with EcoHIV

Following the detection of increased microglia activation, we then explored if this effect was accompanied by increased astrocyte reactivity. To evaluate the status of astroglial activation following EcoHIV infection in WT and TREM2^R47H^ mice, we immunostained vibratome sections of mouse brains with antibody for GFAP using NovaRed for visualization in the HC. The areas analyzed included the following hippocampal sub-regions: CA1, CA2/3, and DG. In the CA1 region of the HC, the GFAP signal was less intense in both saline-treated and EcoHIV-treated WT mice compared to EcoHIV-treated TREM2^R47H^ mice (Figure 3A). The quantification of relative optical density in the CA1 hippocampal sub-region of EcoHIV-treated TREM2^R47H^ mice was ~249% greater than saline-treated WT controls (Figure 3B). EcoHIV-treated TREM2^R47H^ mice had ~282% greater GFAP signal intensity in the CA1 hippocampal sub-region compared to EcoHIV-treated WT mice (Figure 3B). In the CA2/3 region of the HC, GFAP signal intensity was more intense in EcoHIV-treated TREM2^R47H^ mice compared to WT and saline-treated TREM2^R47H^ mice (Figure 3C). Quantification of relative optical density showed ~252% and ~303% greater GFAP signal in the CA2/3 hippocampal sub-region of EcoHIV-treated TREM2^R47H^ mice compared to saline-treated WT and EcoHIV-treated WT mice, respectively (Figure 3D). In the DG region of the HC, GFAP signal was less intense in both saline-treated and EcoHIV-treated WT mice when compared to EcoHIV-treated TREM2^R47H^ mice (Figure 3E). The quantification of corrected optical density in the DG hippocampal sub-region showed ~263% greater GFAP signal intensity in EcoHIV-treated TREM2^R47H^ mice compared to saline-treated WT mice (Figure 3F). EcoHIV-treated TREM2^R47H^ mice also showed ~402% greater GFAP signal in the DG hippocampal sub-region compared to EcoHIV-treated WT mice (Figure 3F). Collectively, EcoHIV infection significantly increased astrocyte reactivity in TREM2^R47H^ mice compared to WT mice. Specifically, an elevated GFAP signal in the hippocampal sub-regions of EcoHIV-treated TREM2^R47H^ mice indicates heightened astrocyte activation associated with the TREM2 R47H gene variant following EcoHIV infection.

### 3.4. The Relationship between TREM2 and Age Is Differentially Affected by Cannabis and HIV

Having established that HIV modulates TREM2 levels and increases glial activity in mice, we further investigated this relationship using ex vivo cultured MDMs. First, we measured TREM2 mRNA in MDMs from PWH and PWoH with varying age and cannabis-use frequency (e.g., naïve, moderate, daily). To begin, correlation analyses were performed to assess the relationship between age and TREM2 mRNA levels. In PWoH without cannabis use, we detected a significant positive correlation between TREM2 mRNA and age (*p* = 0.005; r^2^ = 0.752) (Figure 4A). In PWoH with cannabis use, no significant positive correlation was observed between TREM2 mRNA and age (*p* = 0.243; r^2^ = 0.189) (Figure 4B). Additionally, no significant correlation was observed between TREM2 mRNA and age in PWH that do not use cannabis (*p* = 0.330; r^2^ = 0.047) (Figure 4C). Contrastingly, in PWH with cannabis use, we detected a significant inverse correlation between TREM2 mRNA and age (*p* = 0.009; r^2^ = 0.364) (Figure 4D). These correlation analyses show that TREM2 mRNA levels in MDMs are associated with age and HIV status, with the relationship varying depending on cannabis use. Specifically, these data show a significant positive correlation between TREM2 mRNA and age in PWoH without cannabis use, and this correlation is markedly weaker in PWoH with cannabis use. No such correlation is observed in PWH without cannabis use; however, a significant inverse correlation between TREM2 mRNA levels and age is observed in PWH with cannabis use. Altogether, regulation of TREM2 expression in ex vivo cultured MDMs involves a complex interaction age, cannabis use, and HIV status.

### 3.5. Cannabis Use and HIV Status Differentially Modulate TREM2 Expression in Monocyte-Derived Macrophages Treated with CBD

To further assess the effects of cannabinoids on TREM2 mRNA, MDMs from HIV− and HIV+ Naïve, Moderate, and Daily cannabis users were treated with CBD in the presence of the inflammatory cytokine IL1Β. In HIV− Naive MDMs (Figure 5A), IL1B treatment significantly decreased TREM2 expression by 33.7% compared to vehicle, while CBD treatment resulted in a 33.3% increase. Cotreatment with CBD and IL1B (C + I) increased TREM2 expression by 29.9% relative to IL1B alone. In HIV− Moderate MDMs (Figure 5B), IL1B significantly decreased TREM2 expression by 33.9%, whereas CBD increased TREM2 expression by 34.8% compared to vehicle. Compared to IL1B alone, C + I cotreatment resulted in a 30.8% increase in TREM2 expression. In HIV− Daily MDMs (Figure 5C), IL1B treatment decreased TREM2 expression by 34.8% relative to vehicle controls. In contrast, CBD alone resulted in a 7.5% increase, relative to vehicle, while C + I resulted in a 15.2% increase relative to IL1B treatment. In HIV+ Naive MDMs (Figure 5D), IL1B treatment decreased TREM2 expression by 33.4% compared to vehicle controls, while CBD treatment resulted in a 15.0% increase in TREM2 expression. Cotreatment with C + I increased TREM2 expression by 27.6% compared to IL1B alone. For HIV+ Moderate MDMs (Figure 5E), IL1B decreased TREM2 expression by 32.4% compared to vehicle, while CBD only increased TREM2 expression by 6.2%. The C + I cotreatment resulted in a 7.7% increase in TREM2 expression relative to IL1B alone. In HIV+ Daily MDMs (Figure 5F), IL1B treatment decreased TREM2 expression by 33.2% relative to vehicle controls. CBD treatment resulted in a 3.3% increase in TREM2 expression compared to vehicle, while C + I cotreatment increased TREM2 expression by 25.5% compared to IL1B treatment alone. Across all conditions, these expression analyses show that IL1B treatment consistently reduced TREM2 expression. Co-treatment with CBD and IL1B partially reverses the IL1B-induced reduction in TREM2 expression in both HIV− and HIV+ MDMs, though the response is less robust in HIV+ MDMs. Taken together, differences in TREM2 expression based on cannabis-use frequency under these conditions highlight the modulatory effects of CBD on TREM2′s response to inflammatory stimuli.

To further investigate the mechanisms underlying CBD’s modulation of TREM2 expression in response to inflammatory stimuli, we utilized AM 251, AM 630, GW 6471, and GW 9662 to target cannabinoid receptors and PPAR pathways in MDMs. When comparing HIV− and HIV+ MDMs in the absence of cannabis use (Figure 5G), the reduction in TREM2 expression due to IL1B was more pronounced in HIV+ MDMs (42.8%) than in HIV− MDMs (24.8%). Moreover, CBD treatment significantly increased TREM2 expression in HIV− MDMs by 41.6% compared to only 0.9% in HIV+ MDMs. Differences were also noted in response to AM 630 and GW 6471 combined with C + I, with HIV− MDMs showing significantly higher TREM2 expression (23.6% and 32.6%, respectively) compared to HIV+ MDMs. When comparing HIV− and HIV+ MDMs in the presence of cannabis use (Figure 5H), the reduction in TREM2 expression due to IL1B was 20.5% greater in HIV+ MDMs than in HIV− MDMs. Cotreatment with C + I led to a significantly larger decrease in TREM2 expression in HIV+ MDMs (28.5%) compared to HIV− MDMs (0.6%). When treated with AM 630, GW 6471, or GW 9662 in combination with C + I, TREM2 expression in HIV+ MDMs compared to HIV− MDMs was significantly greater, by 24.2%, 17.5%, and 38.7%, respectively. Overall, we observed key differences in the effects of AM 630 and GW 6471 on TREM2 expression between MDMs from non-cannabis users and cannabis users, with distinct outcomes observed in HIV− and HIV+ groups. In non-cannabis users, C + I cotreatment with AM 630 or GW 6471 revealed significant reductions in TREM2 expression in HIV+ MDMs relative to HIV− MDMs. In cannabis users, however, AM 630 and GW 6471 treatments in the presence of C + I resulted in significantly increased TREM2 expression in HIV+ MDMs relative to HIV− MDMs. These opposing findings further underscore the importance of considering both HIV status and cannabis use when evaluating factors that influence the immunomodulatory role of TREM2 in MDMs. Nevertheless, these results suggest that modulation of IL1B-induced suppression of TREM2 with CBD can be significantly influenced by CB2 and PPARα signaling pathways.

### 3.6. Cannabis Use in PWH Differentially Alters the Gene Expression Profile of MDMs

To continue examining the associations between HIV status, cannabis use, and TREM2-related gene expression, RNA was sequenced from MDMs generated from the following four groups: HIV− Naïve, HIV+ Naïve, HIV+ Moderate, and HIV+ Daily. Volcano plots with corresponding gene ontology (GO) terms depict changes in the genetic profile of HIV+ Naïve MDMs. Relative to HIV− Naïve, HIV+ Naïve MDMs displayed an altered transcriptomic profile with the downregulation of genes such as *TMEM176B*, *CHIT1*, *SERPINF1*, *TMEM176A*, and *CXCL10,* while upregulation was observed in genes such as *DPYSL3*, *ITGB3*, *GREM1*, *TREM1*, and *MMP12* (Figure 6A; fold change > 2, *p* adj < 0.05). Reversed gene expression patterns relative to HIV+ Naïve were seen in HIV+ Moderate MDMs, including the upregulation of genes such as *STAB1*, *TMEM176B*, *CHIT1*, *SERPINF1*, and *TMEM176A*, while downregulated genes included *ITGB3*, *DPYSL3*, *CXCL5*, *GREM1*, and *CCL15* (Figure 6B). Relative to HIV+ Naïve, similar differences in gene expression patterns were seen in HIV+ Daily MDMs, including the upregulation of *STAB1*, *TMEM176B*, *CXCL8*, *SERPINF1*, and *TMEM176A* as well as the downregulation of *ITGB3*, *VSIG4*, *GREM1*, *MMP12*, and *CXCL16* (Figure 6C). Immune and inflammatory processes were among the top identified biological processes among all cannabis backgrounds (Figure 6A–C). To reduce the dimensionality of RNA-sequencing data, a principal component analysis (PCA) plot was generated to visualize patterns of variation among CBD-treated MDMs from HIV+ Naïve, HIV+ Moderate, and HIV+ Daily backgrounds (Figure 6D). The two principal components (PC1 and PC2), representing 50% and 46% of the sample variance respectively, illustrate distinct gene expression profiles associated with CBD-treated MDMs from different cannabis-use patterns in PWH (Figure 6D). Additionally, a heat map was generated to further visualize the differences in expression of TREM2-related genes in MDMs from HIV− Naïve, HIV+ Naïve, HIV+ Moderate, and HIV+ Daily cannabis-use backgrounds (Figure 6E). Further transcriptomic analyses revealed differences in differentially expressed genes, such as CHIT1, SMAD3, ZAP70, TREM1, and VSIG4, in vehicle-treated MDMs from different backgrounds (Figure 6F). MDMs from HIV+ Moderate or HIV+ Daily cannabis-use backgrounds showed significant upregulation of the genes CHIT1, SMAD3, and ZAP70 as compared to HIV− Naïve controls, whereas the same genes were significantly downregulated in MDMs from HIV+ Naïve backgrounds as compared to HIV− Naïve controls (Figure 6F). Contrastingly, TREM1 and VSIG4 gene expression in MDMs from HIV+ Moderate and HIV+ Daily cannabis-use backgrounds were not altered significantly compared to HIV− negative controls. However, TREM1 and VSIG4 gene expression was significantly upregulated in MDMs from HIV+ Naïve cannabis-use backgrounds compared to HIV− negative control (Figure 6F). To further validate the gene expression findings from transcriptomic analyses, RT-qPCR analyses was performed with additional donor samples which revealed comparable patterns in relative gene expression for the genes CHIT1, SMAD3, and VSIG4, (Figure 6G). MDMs from HIV+ Moderate and HIV+ Daily cannabis-use backgrounds had ~101% and ~150% increased expression of CHIT1, respectively, relative to MDMs from HIV+ Naïve backgrounds (Figure 6G). Relative to HIV+ Naïve controls, SMAD3 gene expression increased by ~47% and ~161% in MDMs from HIV+ Moderate and HIV+ Daily cannabis-use backgrounds, respectively. There were no significant differences in the relative expression of ZAP70 and TREM1; however, MDMs from HIV+ Daily backgrounds had a ~112% reduction in relative gene expression of VSIG4 compared to MDMs from HIV+ Naïve backgrounds (Figure 6G). Overall, these findings provide additional implications for understanding how cannabis use and HIV status affect the gene expression profile of MDMs, particularly in the context of TREM2-related gene expression following CBD treatment. The contrasting expression patterns between HIV+ Naïve and HIV+ Moderate or HIV+ Daily MDMs illustrate how cannabis use may suppress some inflammatory pathways while enhancing others. Nevertheless, we show similar gene expression patterns in additional MDMs which substantiate CBD’s impact on TREM2-related gene expression across a broader population.

### 3.7. CBD Increases Membrane-Bound TREM2 and Reduces sTREM2 in MDMs from PWH

To determine the effect of CBD on the cleavage of TREM2, membrane-bound TREM2 was visualized via ICC on fixed MDMs and sTREM2 was measured via ELISA using media collected from treated MDMs. The intensity of membrane-bound TREM2 was significantly reduced in IL1B-treated MDMs (Figure 7A). The quantification of relative TREM2 intensity revealed a ~38% reduction in TREM2 signal from IL1B-treated MDMs relative to vehicle-controls (Figure 7B). The cotreatment with CBD and IL1B resulted in a significant increase in TREM2 signal relative to IL1B alone and a ~16% decrease in TREM2 signal relative to vehicle controls (Figure 7B). Moreover, IL1B-treated MDMs had ~45% increased relative absorbance of sTREM2 compared to vehicle controls, and CBD-treated MDMs had the lowest relative absorbance of sTREM2, compared to vehicle controls (Figure 7C). Although non-significant, cotreatment with CBD and IL1B showed a ~18% decrease in relative absorbance of sTREM2 compared to media collected from IL1B-treated MDMs (Figure 7C). All in all, these data highlight the significant impact of CBD on TREM2 dynamics, specifically its ability to counteract the IL1B-induced cleavage of TREM2 and reduce the levels of pro-inflammatory sTREM2. These results demonstrate the potential for CBD to counteract the elevated levels of sTREM2 that contribute to pathological inflammation.

## 4. Discussion

In this study, we investigated the relationship between HIV and TREM2 expression and the impact of cannabis use on TREM2 expression in PWH. This is the first study to show decreased total TREM2 expression in response to EcoHIV infection, supporting previous work which showed decreased TREM2 in membrane-enriched fractions of brain homogenates from PWH with HAND [16]. This work also shows that EcoHIV infection is associated with increased microglial activation in TREM2 R47H mice, as evidenced by differential IBA1 expression in mouse brains. Consistent with these findings, other studies have shown that alterations in TREM2 expression can modulate microglial activation and influence cognitive function in various disease models [65,66]. Importantly, this study reports altered levels of TREM2 and related gene expression in myeloid lineage cells from a cohort of PWH with varying cannabis-use frequencies. Lastly, CBD increased TREM2 RNA and membrane-bound TREM2 in monocyte-derived macrophages. These findings may be increasingly important as PWH live longer on ART and may be at increased risk for developing age-related neurodegenerative diseases. Due to potential compounding effects of HIV and aging on the mechanisms of neural insult such as inflammation [67,68], novel therapeutic strategies are needed to stop or prevent HIV-associated neuropathogenesis in an aging population.

PWH carrying gene variants of TREM2, such as R47H TREM2, have been associated with an increased risk of developing AD [69,70,71,72]. Moreover, HIV-induced inflammation is a possible mechanism for AD-like pathogenesis and viral infection is suspected as a possible etiology of sporadic AD [73,74]. Postmortem brain tissues from PWH present many inflammatory markers that occur in AD [75,76,77]. Compared to the general population, older PWH are also at risk for age-associated neurodegeneration including amnestic mild cognitive impairment (aMCI) [78]. The EcoHIV model is well-recognized for its utility in representing the acute phase of HIV infection in mice, characterized by initial viral replication and immune response, including cytokine production [13,79]. While this model simulates an acute infection, it is important to note that EcoHIV does invade the brain, albeit at low levels, leading to cytokine responses that are consistent with mild neurocognitive impairment (NCI) observed in humans [80]. These findings are in line with EcoHIV inducing a reduction in TREM2 protein levels in mouse brains and increasing microgliosis in TREM^2R47H^ mice in this study. Furthermore, EcoHIV-infected TREM^2R47H^ mice performed worse in behavioral testing for learning and memory which is indicative of memory deficits, a defining characteristic of aMCI [81]. It is plausible that genetic mutations and exposure to viral infection may initiate inflammatory cascades upstream of AD pathogenesis, much like the potential mechanisms controlling TREM2 dysfunction [82,83].

Unlike microglia and perivascular macrophages which do not infiltrate the brain by crossing the blood–brain barrier [10,21], MDMs are thought to transport HIV into the brain soon after initial infection [11,84]. In addition to increased levels of Aβ [77,85,86], HIV infection of the brain is associated with inflammatory cytokines such as interleukin (IL)-1β [16,87,88,89]. TREM2 exerts its anti-inflammatory effects primarily through modulation of brain macrophage function and the immune response within the CNS. Cannabis contains hundreds of other phytocannabinoids, terpenoids, and polyphenols that may exert immunomodulatory functions through TREM2-related pathways [90]. Indeed, recent evidence suggests that certain plant-derived compounds exert anti-inflammatory actions on microglial cells via the TREM2 signaling pathway [91]. Thus, cannabinoids may have anti-inflammatory neuroprotective benefit for AD and HIV NCI. Notably, cannabis use is associated with reduced inflammation and lower likelihood of cognitive impairment in PWH [53,55,56,92,93]. It was for these reasons that we chose to investigate TREM2 expression in MDMs from a cohort of PWH compared to PWoH. Our findings are consistent with previous studies that showed alterations in TREM2 levels in the brain and CSF of PWH [16,43] and that these changes may be affected by age. These findings also bolster the case for identifying therapeutic strategies to modulate TREM2 expression to prevent or treat neurodegenerative diseases.

Cannabidiol (CBD) reduces inflammation in myeloid lineage cells exposed to inflammatory stimuli, including HIV, HIV proteins, Aβ, and inflammatory cytokines that are implicated in both HIV NCI and AD [94,95,96,97]. THC has shown neuroprotective properties in mouse models for HIV NCI and AD [93,96,98,99,100,101,102,103,104,105]. The data presented here indicate cannabinoids may promote TREM2 expression and function. For example, the differences in age-associated changes in TREM2 expression between PWH and PWoH and the association with cannabis patterns may support a role for cannabinoids in modulating the TREM2 pathway. Specifically, cannabis use was linked to lower TREM2 mRNA levels in older individuals. Cannabis may also modulate many MDM-related genes altered by HIV infection. The observed HIV-associated changes in the expression of immunomodulatory genes, including CHIT1 [106], SMAD3 [107], ZAP70 [108], TREM1 [109,110], and VISG4 [111], were differentially impacted by cannabis use. RNA-sequencing analyses revealed additional differences in the gene expression profile of MDMs from moderate or daily cannabis-use backgrounds. This is in line with recent findings that illustrate that daily cannabis use is associated with lower inflammation in the CNS of PWH [51,52,54,56,112].

The opposing effects of IL1Β and CBD on sTREM2 levels and TREM2 protein expression suggests a complex interplay between inflammatory stimuli and cannabinoid signaling pathways. These results indicate that CBD may ameliorate inflammation in part by maintaining TREM2 expression and reducing sTREM2, potentially through the modulation of specific signaling pathways involved in TREM2 processing. Inhibiting the activity of ADAM10 and ADAM17, which are key proteases responsible for cleaving TREM2 into its soluble form [113], can occur at the level of intracellular localization [114]. This selective inhibition could result in the preservation of membrane-bound TREM2, which is crucial for maintaining microglial homeostasis and promoting phagocytosis of amyloid-beta plaques [115]. In contrast, the lack of a protective effect on soluble TREM2 levels may indicate that CBD does not influence the subsequent steps involved in TREM2 shedding or may even promote alternative cleavage pathways that bypass the inhibition of ADAM proteases. Despite its involvement in several cellular processes [116], it remains unclear if CBD selectively inhibits the intracellular mechanisms involved with TREM2 cleavage. Additional research is not only necessary to understand the complex molecular makeup of cannabis, but also is imperative to elucidate the potential underlying molecular mechanisms shared by TREM2 and cannabis-derived anti-inflammatory mediators like CBD. Nevertheless, these findings are in line with emerging evidence suggesting bidirectional interactions between cannabinoids and neuroinflammation [81].

While the endocannabinoid system has been shown to regulate immune cell function via the cannabinoid type-2 (CB2) receptor, the specific mechanisms underlying the effects of cannabis on TREM2 expression remain unclear and warrant further investigation. Nevertheless, CBD has been shown to modulate microglial activation in various disease models, including HIV-associated and Aβ-induced neuroinflammation [58,90,117,118]. The anti-inflammatory actions of cannabis are largely attributed to cannabinoids which can act on multiple targets within the endocannabinoid system, including the cannabinoid type-2 (CB2) receptor [59,119]. However, CBD has also been shown to exert its effects on microglia through other signaling pathways [120]. The pharmacological agents AM 251, AM 630, GW 6471, and GW 9662 have been utilized to elucidate the roles of cannabinoid receptors and peroxisome proliferator-activated receptors (PPARs) in immunomodulation. AM 251, a selective CB1 receptor antagonist, and AM 630, a CB2 receptor antagonist, are commonly used to block cannabinoid receptor activity. Studies indicate that blocking CB1 and CB2 receptors can influence macrophage polarization and immune response. For example, AM 630 has been shown to alter the anti-inflammatory effects on MDMs, suggesting a role for CB2 in macrophage-mediated immune responses [121]. GW 6471, a PPARα antagonist, and GW 9662, a PPARγ antagonist, are used to investigate the involvement of PPARs in immune cell function. PPARγ, in particular, is known for its role in regulating macrophage activation and inflammation. The inhibition of PPARγ by GW 9662 was observed to impact macrophage differentiation and inflammatory cytokine production, indicating its regulatory function in immune responses [122]. These reported anti-inflammatory effects of PPARγ are consistent with our findings that GW9662 blocked the effect of CBD to increase TREM2 mRNA in IL1β-treated MDMs. Indeed, studies have shown that disrupted PPARγ signaling may impact TREM2 function and microglial immunometabolic phenotype [123]. Nevertheless, the anti-inflammatory activity of CBD is consistent with our studies in which the treatment of human MDMs with CBD consistently increased TREM2 mRNA regardless of HIV status or cannabis-use frequency. Moreover, CBD treatment mitigated the IL1B-induced downregulation of TREM2, which may reduce the impact of inflammatory stimuli in the brain [124]. These findings suggest that the immune-modulatory effects of CBD and other cannabinoids are influenced by prior cannabis use and underscore the importance of considering cannabis-use history in studies evaluating the therapeutic potential of cannabinoids in individuals with HIV.

While our study provides valuable insights into the role of TREM2 and the potential impact of cannabis on neuroinflammation in HAND, it is essential to acknowledge several limitations that may affect the generalizability and interpretation of our findings. Firstly, our study relied on a limited number of donors, which may not fully represent the diversity and variability within the population of people living with HIV. Additionally, the age of donors may not encompass the full spectrum of age-related changes that could influence TREM2 expression and neuroinflammatory processes. Moreover, while our analysis revealed alterations in TREM2 expression and neuroinflammatory markers in an EcoHIV-infected mouse model, it is crucial to recognize that this model lacks key components of HIV neuropathogenesis, such as the gp120 protein. Furthermore, our study did not consider the potential effects of antiretroviral therapy (ART) in the mouse model which could significantly influence TREM2 expression, neuroinflammation, and cognition. This line of reasoning may explain the lack of negative effects from the NOR results in EcoHIV-treated mice. We also recognize the critical importance of viral load as a variable in HIV pathogenesis, particularly its impact on neuroinflammation and cognitive function, and future studies should aim to report viral load values when possible. Moreover, future studies should incorporate additional markers to better distinguish between microglial subtypes and to differentiate microglia from peripheral macrophages, thereby providing a more nuanced understanding of microglial function and activation in the context of HIV-associated neuroinflammation. Lastly, our study did not explore the effects of minor cannabinoids that are present in cannabis, and these may be responsible for observed associations with reduced inflammation and neuroprotection in PWH. Therefore, while our findings provide valuable insights, further research with larger sample sizes, diverse populations, consideration of age-related changes, inclusion of relevant HIV components, and exploration of therapeutic interventions is warranted to validate and extend our findings.

## 5. Conclusions

In summary, our study provides novel insights into the intricate relationship between TREM2, neuroinflammation, and cognitive function in the context of HIV infection and cannabis use. First, findings from this murine model highlight the importance of considering the interaction of HIV and TREM2 dysfunction which may further exacerbate the cognitive and behavioral deficits associated with chronic neuroinflammation. Secondly, our ex vivo findings indicate that cannabis use and HIV may modulate TREM2 and related gene expression in myeloid cells. Lastly, CBD’s anti-inflammatory potential to lessen neuroinflammation is linked to changes in both TREM2 mRNA and protein expression. Altogether, results from this study underscore the potential of TREM2 as a therapeutic target for the treatment of HAND. Further research is warranted to elucidate the specific mechanisms underlying these interactions and to explore potential therapeutic strategies targeting TREM2 and cannabinoid signaling pathways in neuroinflammatory diseases.

## Figures and Tables

**Figure 1 viruses-16-01509-f001:**
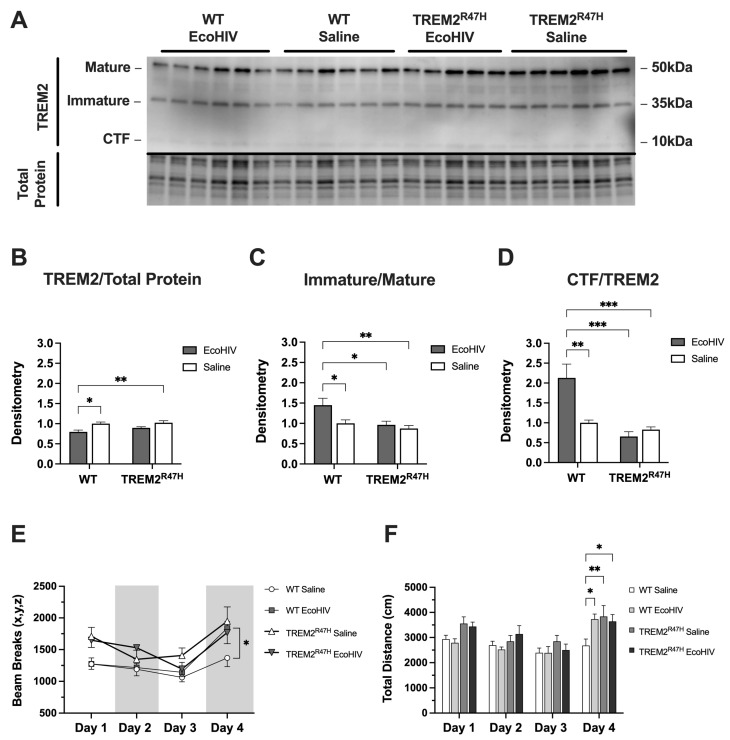
EcoHIV reduces levels of TREM2 and alters memory in WT and TREM2^R47H^ mice. (**A**) Detection of the mature, immature, and CTF isoforms for TREM2 in wild type (WT) and TREM2^R47H^ mice treated with EcoHIV or saline. Densitometry ratios for (**B**) total TREM2, (**C**) immature/mature TREM2 isoforms, and (**D**) CTF isoform/total TREM2, all normalized to total protein band density. (**E**) Total activity measurements represented as beam breaks over four days in WT and TREM2^R47H^ mice treated with EcoHIV or saline. (**F**) Total distance traveled during behavioral testing in WT and TREM2^R47H^ mice treated with EcoHIV or saline. Day 4 of behavioral tests occurred 72 h after mice were tested for three consecutive days (days 1–3). Data presented as mean ±  SEM and analyzed using two-way ANOVA with Holm–Sidak’s multiple comparisons tests; *n* = 4–6 per condition; * *p* < 0.05, ** *p* < 0.01, *** *p* < 0.001. Abbreviations: CTF, carboxy terminal fragment.

**Figure 2 viruses-16-01509-f002:**
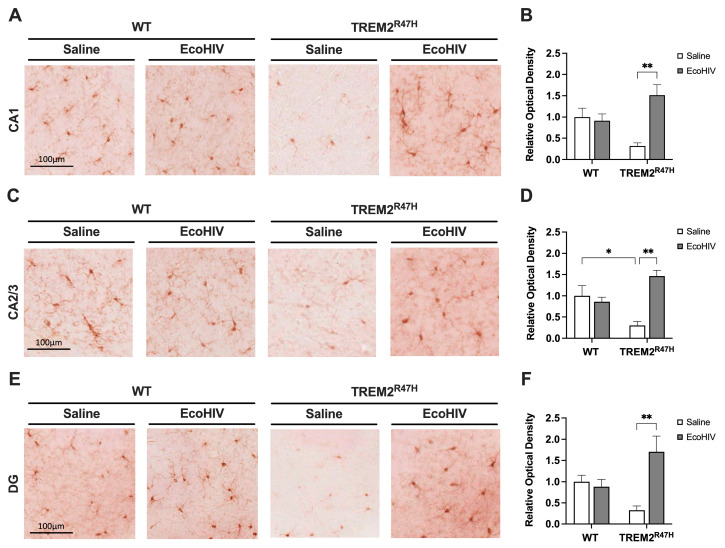
EcoHIV increases hippocampal IBA1 expression in TREM2^R47H^ mice. (**A**) IBA1-immunostained vibratome sections in the CA1 sub-region of the hippocampus from wild type (WT) mice and TREM2^R47H^ mice treated with saline or EcoHIV. (**B**) Relative optical density quantification of IBA1 in CA1 sections. (**C**) IBA1-immunostained vibratome sections in the CA2/3 sub-region of the hippocampus of WT mice and TREM2^R47H^ mice treated with saline or EcoHIV. (**D**) Relative optical density quantification of IBA1 in CA2/3 sections. (**E**) IBA1-immunostained vibratome sections in the DG sub-region of the hippocampus of WT mice and TREM2^R47H^ mice treated with saline or EcoHIV. (**F**) Relative optical density quantification of IBA1 in DG sections. Scale bars represent 100 μm. Data presented as mean ± SEM normalized to WT-saline controls and analyzed using two-way ANOVA with Holm–Sidak’s multiple comparisons tests; *n* = 4–6 per condition; statistical significance was determined at * *p* < 0.05, ** *p* < 0.001. Abbreviations: IBA1, ionized calcium-binding adapter molecule 1; CA, cornu ammonis; DG, dentate gyrus.

**Figure 3 viruses-16-01509-f003:**
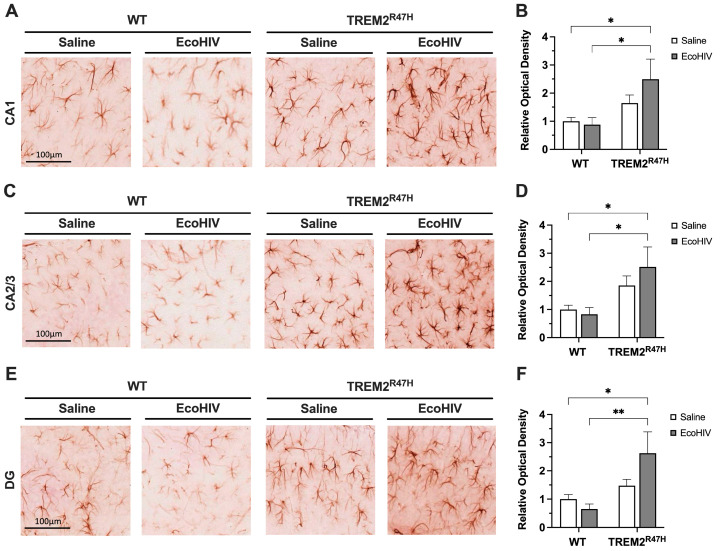
Hippocampal GFAP expression is increased in TREM2^R47H^ mice infected with EcoHIV. (**A**) GFAP -immunostained vibratome sections in the CA1 sub-region of the hippocampus from wild type (WT) mice and TREM2^R47H^ mice treated with saline or EcoHIV. (**B**) Relative optical density quantification of GFAP in CA1 sections. (**C**) GFAP-immunostained vibratome sections in the CA2/3 sub-region of the hippocampus of WT mice and TREM2^R47H^ mice treated with saline or EcoHIV. (**D**) Relative optical density quantification of GFAP in CA2/3 sections. (**E**) GFAP-immunostained vibratome sections in the DG sub-region of the hippocampus of WT mice and TREM2^R47H^ mice treated with saline or EcoHIV. (**F**) Relative optical density quantification of GFAP in DG sections. Scale bars represent 100 μm. Data presented as mean ± SEM normalized to WT-saline controls and analyzed using two-way ANOVA with Holm–Sidak’s multiple comparisons tests; *n* = 4–6 per condition; statistical significance was determined at * *p* < 0.05, ** *p* < 0.001. Abbreviations: GFAP, glial fibrillary acidic protein; CA, cornu ammonis; DG, dentate gyrus.

**Figure 4 viruses-16-01509-f004:**
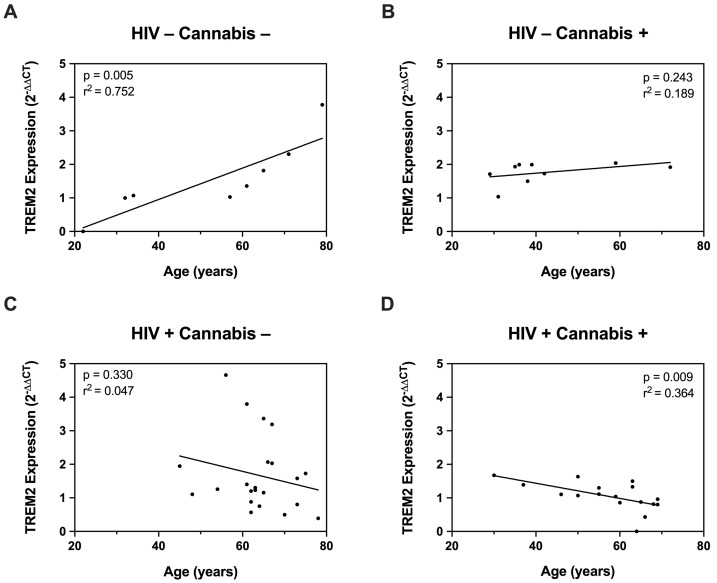
The relationship TREM2 and age is differentially affected by cannabis and HIV. (**A**) Correlation plot for TREM2 expression in MDMs versus age in PWoH with no cannabis use (*n* = 8). (**B**) Correlation plot for TREM2 expression in MDMs versus age in PWoH that use cannabis. (*n* = 9). (**C**) Correlation plot for TREM2 expression in MDMs versus age in PWH with no cannabis use (*n* = 22). (**D**) Correlation plot for TREM2 expression in MDMs versus age in PWH that use cannabis (*n* = 17). Data presented as mean ± SEM relative fold change in TREM2 using the comparative CT method (2^−ΔΔCq^) and normalized to ACTB. Each point represents individual expression values in MDMs collected from unique donors. All correlation plots analyzed via simple linear regression. Individual significance (*p*-value) and goodness of fit coefficient (r^2^) determined for each correlation and listed within plots. *n* = 8–22 per plot; statistical significance was determined at *p* < 0.05.

**Figure 5 viruses-16-01509-f005:**
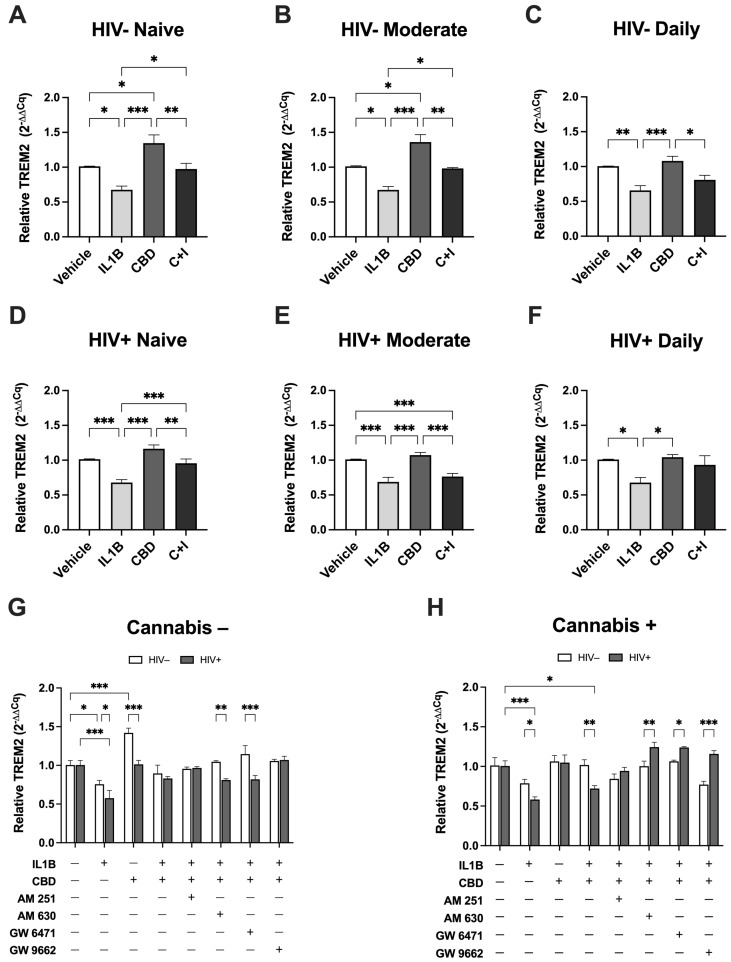
Cannabis use and HIV status differentially modulate TREM2 expression in monocyte-derived macrophages treated with CBD. (**A**) Relative TREM2 expression in HIV− Naïve (*n* = 10), (**B**) HIV− Moderate (*n* = 3), (**C**) HIV− Daily (*n* = 8), (**D**) HIV+ Naïve (*n* = 22), (**E**) HIV+ Moderate (*n* = 13), and (**F**) HIV+ Daily (*n* = 7) MDMs treated with IL1B (20 ng/mL), CBD (30 µM), or cotreatment with CBD and IL1B (C + I). (**G**,**H**) Relative TREM2 expression after treatment with IL1B, CBD, or C + I alone and in combination with cannabinoid PPAR antagonists in (**G**) non-cannabis-user MDMs and (**H**) cannabis-user MDMs. Data presented as mean ±  SEM relative expression in TREM2 normalized to vehicle controls using ACTB. Data analyzed using one-way (**A**–**F**) and two-way (**G**,**H**) ANOVA with Holm–Sidak’s multiple comparisons tests. Statistical significance was determined at * *p* < 0.05, ** *p* < 0.01, *** *p* < 0.001. All inhibitor concentrations were 10 µM. AM 251, CB1 antagonist; AM 630, CB2 antagonist; GW 6471, PPARα antagonist; GW 9662, PPARγ antagonist.

**Figure 6 viruses-16-01509-f006:**
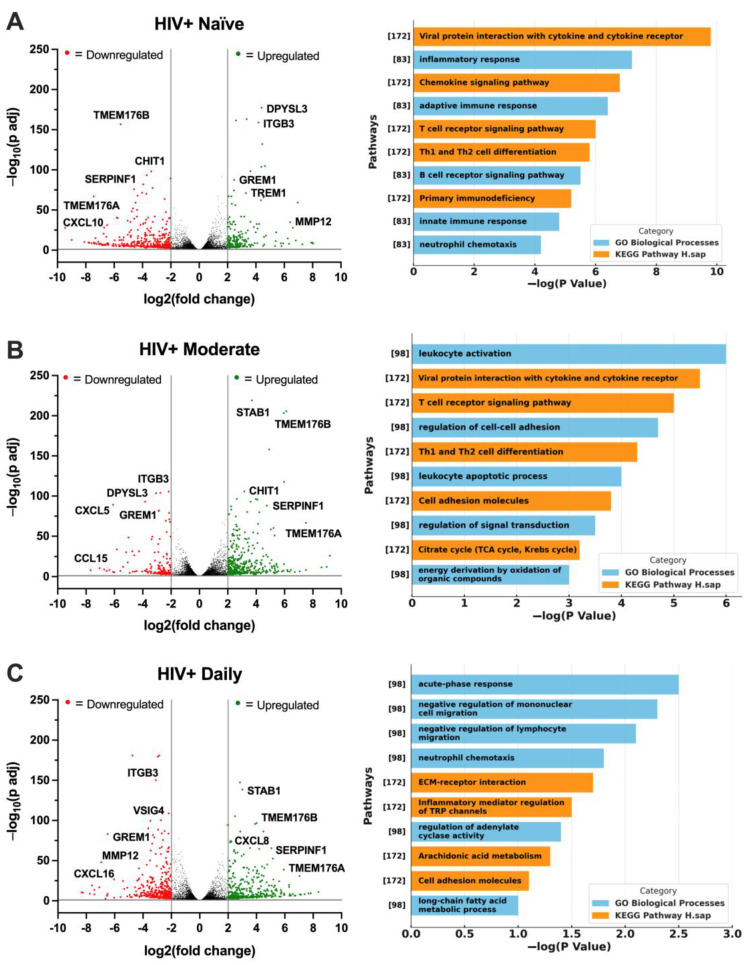
Cannabis use in PWH differentially alters the gene expression profile of MDMs. (**A**–**C**) Volcano plots with corresponding gene ontology (GO) term plots for (**A**) HIV+ Naïve, (**B**) HIV+ Moderate, and (**C**) HIV+ Daily MDMs. (**D**) Principal component analysis (PCA) plot displaying sample variance among MDMs from PWH with varying cannabis-use frequency. (**E**) Heat map of TREM2-related and other relevant genes in MDMs treated with CBD (30 µM), IL1B (20 ng/mL), or C + I. (**F**) Differentially expressed genes from RNA-seq analyses. (**G**) RT-qPCR analyses of differentially expressed genes using additional donor samples. Data presented as mean ±  SEM analyzed using one-way ANOVA with Holm–Sidak’s multiple comparisons tests. Statistical significance was determined at * *p* < 0.05, ** *p* < 0.01, *** *p* < 0.001. Abbreviations: *CHIT1*, Chitinase 1; *SMAD3*, SMAD Family Member 3; *ZAP70*, Zeta Chain of T Cell Receptor Associated Protein Kinase 70; *TREM1*, Triggering Receptor Expressed on Myeloid Cells 1; *VSIG4*, V-Set and Immunoglobulin Domain Containing 4.

**Figure 7 viruses-16-01509-f007:**
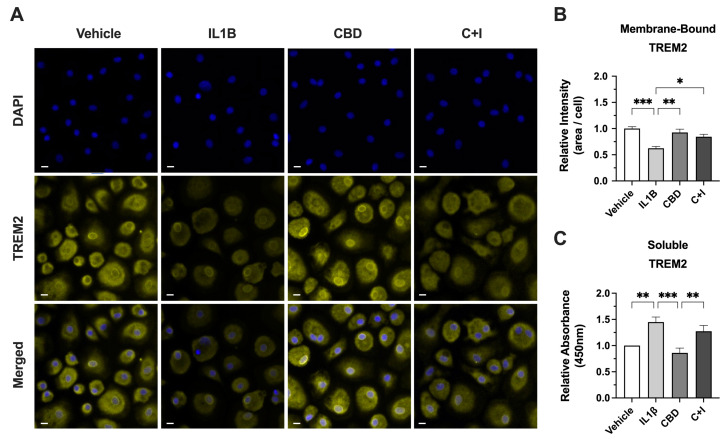
CBD increases membrane-bound TREM2 and reduces sTREM2 in MDMs from PWH. (**A**) Representative images of MDMs from HIV+ Naïve background immunostained for membrane-bound TREM2 following 24 h treatment with IL1Β (20 ng/mL), CBD (30 µM), or C + I. (**B**) Relative TREM2 intensity for immunostained MDMs treated with IL1Β, CBD, or C + I. (**C**) Relative absorbance for sTREM2 measured via ELISA in media collected from cultured MDMs treated with IL1Β, CBD, or CBD + IL1B (C + I). Scale bars represent 10 μM. Data normalized to vehicle controls and presented as mean ± SEM and analyzed using one-way ANOVA with Holm–Sidak’s multiple comparisons tests; *n* = 3–4 per condition; * *p* < 0.05, ** *p* < 0.01, *** *p* < 0.001.

**Table 1 viruses-16-01509-t001:** Demographic, clinical and cannabis-use characteristics of study population.

Variable	HIV− (*n* = 17)	HIV+ (*n* = 39)	Overall (N = 56)
Sex (M/F)	13/4	36/3	49/7
Age (years ± SEM)	48.5 ± 3.9	60.7 ± 2.2	55.7 ± 2.0
Race/ethnicity (% White)	58.8	71.8	67.9
(% Hispanic)	5.9	15.4	12.5
(% Other)	35.3	12.8	19.6
Cannabis user (%Yes)	52.9	40.5	44.9
**Group**	**Age** **(years ± SEM)**	**GDS** **(score ± SEM)**	**Last Cannabis Use** **(Days ± SEM)**
HIV− Naïve (*n* = 8)	53.2 ± 6.5	0.17 ± 0.06	3398.1 ± 2523.2
HIV− Moderate (*n* = 1)	29.0 ± n/a	0.06 ± N/A	30.0 ± N/A
HIV− Daily (*n* = 8)	44.0 ± 4.9	0.27 ± 0.11	0.3 ± 0.2
HIV+ Naive (*n* = 22)	63.7 ± 1.7	0.32 ± 0.07	3173.3 ± 949.3
HIV+ Moderate (*n* = 11)	55.8 ± 3.5	0.42 ± 0.14	826.4 ± 652.9
HIV+ Daily (*n* = 6)	52.9 ± 5.2	0.49 ± 0.28	0.9 ± 0.4

Abbreviations: global deficit score (GDS). Not applicable (N/A) denotes cases where the standard error of the mean (SEM) cannot be calculated due to only one participant in the group.

## Data Availability

All data will be available by reasonable request and will also be deposited with Gene Expression Omnibus.

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
