# Peer review of "Cannabis Use and Cannabidiol Modulate HIV-Induced Alterations in TREM2 Expression: Implications for Age-Related Neuropathogenesis"

_viruses, 2024, doi:10.3390/v16101509_

Round 1
Reviewer 1 Report
Comments and Suggestions for Authors
In this study titled "HIV-induced modulation of TREM2 expression is reversed by cannabidiol: implications for age-related neuropathogenesis in people with HIV", the authors present a clear examination of the relationship between TREM2 and neuroinflammation in the context of HIV-induced neuropathogenesis, and, most crucially, present further proof that cannabidiol (CBD) plays a modulating role in TREM2 expression and can probably inhibit the neuropathogenesis instigated by HIV. The employment of mouse models and samples also of human origin helps in obtaining essential information on the intersection of HIV, neuroinflammation, and cannabinoid therapy. The study is well thought of, and the use of wildtype and TREM2R47H mutant mice, involving monocyte-derived macrophages (MDMs) from humans with or without HIV, makes the study translational.
Here are some comments below:
1. In Fig. 1, There is no indication of the level of significance depicted by means such as *, **, or ***.
2. Fig. 1: The WB samples seem to have inconsistencies in Actin levels, the authors can give more details on how the samples to be balanced and improve the quality of WB.
3. Fig. 2: The authors demonstrated the whole hippocampi using IBA1 immunostaining. Additionally, they provided magnified sections of CA1, CA2/3, and DG. For a better presentation, the effects of boxes on the whole hippocampi photos should be considered in order to indicate the exact places of the area those images shown in magnified sections.
4. Fig. 2: The IBA1 optical density analysis is not match with its the representative image. For example, in fig.2a, the density of IBA1 in the WT-EchoHIV group shows darker than in the saline group. However, the analysis data presents in fig.2b shows no significant difference. Additionally, in fig.2c-d, 2e-f and 2g-h, the presentative images shows not much significant difference of IBA1 optical density among group of TREM2-EcoHIV, WT saline and EcoHIV groups. But, the analysis data show a significantly increase of IBA1 density in Trem2-EcoHIV group compared with other two groups. Could the authors please clarify how the data were analyzed.
5. Figure 3: For the same reason explained in the previous section, just placing boxes on the magnified hippocampi to show where they actually are would really improve the figure’s clarity.
6. Fig. 7A: According to the results, there was no significant difference in the C+I and IL-1β groups. So, perhaps the authors could explain why the CBD, while not being able to protect IL-1β-mediated reduction in soluble Trem2, was able to inhibit IL-1β-induced downregulation of membrane-bound TREM2. Furthermore, the authors might make a discussion on the implications of this difference would be valuable.
7. Fig. 7B: The scale bar is missing.
Author Response
- In Fig. 1, There is no indication of the level of significance depicted by means such as *, **, or ***.
Response:
- Thank you for pointing this out. We will revise Figure 1 to include the appropriate indicators of statistical significance (e.g., *, **, ***) to denote the p-values associated with the differences observed between groups. The figure legend will also be updated to define the significance levels (e.g., *p < 0.05, **p < 0.01, ***p < 0.001) and to specify the statistical tests used.
================================================================================
- Fig. 1: The WB samples seem to have inconsistencies in Actin levels, the authors can give more details on how the samples to be balanced and improve the quality of WB.
Response:
- We appreciate your observation regarding the variability in ACTB levels across the Western Blot samples. We reexamined the data and performed densitometric normalization to total protein to ensure accurate quantification of TREM2 protein. We also include additional detailed methods regarding how samples were balanced to total protein in an effort to improve the reliability and clarity of the data presented, as noted below:
- Page 4, lines 177-178: Average adjusted total band intensities of total TREM2 protein (mature; 50kD + immature; 35kD + CTF; 10kD) were normalized to WT-Saline controls.
================================================================================
- Fig. 2: The authors demonstrated the whole hippocampi using IBA1 immunostaining. Additionally, they provided magnified sections of CA1, CA2/3, and DG. For a better presentation, the effects of boxes on the whole hippocampi photos should be considered in order to indicate the exact places of the area those images shown in magnified sections.
Response:
- We agree that additional clarity is needed indicate the exact regions shown in the magnified sections of Figure 2. The initial whole hippocampi sections were primarily shown to illustrate where the magnified regions are approximately located within the overall hippocampal structure. To illustrate the differences accurately and clearly between representative samples for each area of the hippocampus, we changed the figure to show only the magnified images from CA1, CA2/3, and DG regions.
- These changes are seen in Figure 2, and can be seen on page 14, line 562.
================================================================================
- Fig. 2: The IBA1 optical density analysis is not match with its the representative image. For example, in fig.2a, the density of IBA1 in the WT-EchoHIV group shows darker than in the saline group. However, the analysis data presents in fig.2b shows no significant difference. Additionally, in fig.2c-d, 2e-f and 2g-h, the presentative images shows not much significant difference of IBA1 optical density among group of TREM2-EcoHIV, WT saline and EcoHIV groups. But, the analysis data show a significantly increase of IBA1 density in Trem2-EcoHIV group compared with other two groups. Could the authors please clarify how the data were analyzed.
Response:
- Thank you for highlighting this issue. We re-examined the images and the corresponding data analysis and corrected discrepancies in magnification and resolution among the initial images. Additionally, one data point from the WT-EcoHIV group was removed from all analyses following confirmation that EcoHIV was not detectable in the mouse. As such, we updated the figure and analysis to ensure consistency between the representative visual data and the quantitative findings. We also included an updated detailed description of the relative optical density analysis process in the Methods section, as listed below:
- Page 5, lines 200-205: Average intensity of the immunostaining in areas of interest in the hippocampus (e.g. CA1, CA2/3, DG regions) were corrected for average background levels obtained from control sections processed without primary antibody. Optical density values for both IBA1 and GFAP within each of the regions were then normalized to the respective WT-Saline controls for relative quantification.
================================================================================
- Figure 3: For the same reason explained in the previous section, just placing boxes on the magnified hippocampi to show where they actually are would really improve the figure’s clarity.
Response:
- We again appreciate the reviewer’s insightful suggestion to enhance clarity. To avoid potential confusion arising from minor individual differences in the exact location of the representative magnified images, we have decided to also remove the whole hippocampi images from Figure 3. Instead, we have focused solely on the magnified sections of CA1, CA2/3, and DG, which are now presented with clearer resolutions and annotations.
- The updated Figure 3 can be found on page 15, line 576.
================================================================================
- Fig. 7A: According to the results, there was no significant difference in the C+I and IL-1β groups. So, perhaps the authors could explain why the CBD, while not being able to protect IL-1β-mediated reduction in soluble Trem2, was able to inhibit IL-1β-induced downregulation of membrane-bound TREM2. Furthermore, the authors might make a discussion on the implications of this difference would be valuable.
Response:
- We acknowledge that the differential effects of CBD on soluble versus membrane-bound TREM2 warrant further discussion. While the precise mechanisms remain unclear, it is possible that CBD exerts selective modulation on the pathways involved in the processing and cleavage of TREM2, specifically affecting the maintenance of its membrane-bound form. This selective modulation may be related to the differential signaling pathways activated by membrane-bound versus soluble TREM2, with CBD potentially stabilizing the membrane-bound form through its effects on the cellular microenvironment or proteolytic enzymes involved in TREM2 cleavage.
- To address this, we have expanded the discussion to consider these potential mechanisms and their broader implications, particularly in the context of neuroinflammatory conditions where the balance between membrane-bound and soluble TREM2 might influence disease progression. These additions can be found on page 22, lines 714-728: The opposing effects of IL1Β and CBD on sTREM2 levels and TREM2 protein ex-pression suggests a complex interplay between inflammatory stimuli and cannabinoid signaling pathways. These results indicate that CBD may ameliorate inflammation in part by maintaining TREM2 expression and reducing sTREM2, potentially through modulation of specific signaling pathways involved in TREM2 processing. Inhibiting the activity of ADAM10 and ADAM17, which are key proteases responsible for cleaving TREM2 into its soluble form[115], can occur at the level of intracellular localization[116]. This selective inhibition could result in the preservation of membrane-bound TREM2, which is crucial for maintaining microglial homeostasis and promoting phagocytosis of amyloid-beta plaques[117]. In contrast, the lack of a protective effect on soluble TREM2 levels may indicate that CBD does not influence the subsequent steps involved in TREM2 shedding or may even promote alternative cleavage pathways that bypass the inhibition of ADAM proteases. Despite its involvement in several cellular processes[118], it remains unclear if CBD selectively inhibits the intracellular mechanisms involved with TREM2 cleavage.
================================================================================
- Fig. 7B: The scale bar is missing.
Response:
- Thank you for pointing this out. We have added the missing scale bar in Figure 7B, now Figure 7A, as shown on page 21, line 632. The figure legend was also updated to include information on the scale bars, as shown on page 21, line 638: Scale bars represent 10 μM.
================================================================================
Reviewer 2 Report
Comments and Suggestions for Authors
The study from Avalos et al. offers new insights into the complex interplay between TREM2, neuroinflammation, and cognitive function within the context of HIV infection and cannabis use.
Major concerns
1. Please include in the Introduction important concepts for readers who are not specialists in the subject. These should include the definition and functional characteristics of TREM2 in its mature and immature forms, expression of the CB2 receptor in microglia (not just in macrophages), abbreviations (carboxy-terminal fragment), and their meanings (HAND, DSM, etc.).
2. Have the authors considered that the model EcoHIV used—according to the evaluation timeline—represents an acute infection in the murine model? Additionally, have they considered that despite EcoHIV invading the brain at low levels and inducing cytokine responses in the brain consistent with mild brain disease in humans, these characteristics might limit the interpretation of the results? This is an experimental animal system that may allow modeling host pathophysiological conditions for mild neurocognitive impairment (NCI) induction in HIV-suppressed people. EcoHIV infection causes NCI but not CD4+ T cell depletion or immunodeficiency in mice Please also explain how did the authors defined the EcoHIV dose used.
3. Could the authors provide a brief explanation of the rationale for selecting frontal cortex tissues for the analysis of TREM2 expression? Likewise, please specify the sections of the mouse brain that were analyzed for IBA1 and GFAP immunohistochemistry.
4. In contrast to so-called homeostatic microglial markers, such as transmembrane protein (TMEM-)119 or P2RY12, IBA1 is also expressed by peripheral macrophages. Even though IBA1 is often characterized as a typical microglial activation marker, it cannot be used to distinguish between functional microglial phenotypes. So, do the authors consider that this marker presents limitations in the evaluation conducted?
5. The interpretation (at the end) of the Results sections 3.3, 3.4, 3.5, 3.6, and 3.7 is not included. The authors' interpretation is valuable for readers prior to its discussion.
6. Check this paragraph (section 3.6): "Relative to HIV+ Naïve, MDMs from HIV+ Moderate PWH displayed an altered transcriptomic profile with XXX and YYY genes up- and down-regulated, respectively".
7. Table 1. TBD? please specify.
Only one participant has been included in the group HIVneg/moderate.
8. In Figure 1, the asterisks representing the statistical analysis were not included.
In Figure 4, the number of samples analyzed (points in each graph) does not match the details provided in the figure caption.
9. The authors claim to "corroborate" results found in patients with those observed in the murine model using EcoHIV. ("Our observation of decreased total TREM2 expression in response to EcoHIV infection corroborates previous work which shows decreased TREM2 in membrane-enriched fractions of brain homogenates from PWH with HAND" Discussion section). Can this model truly CORROBORATE what is observed in patients?
10. The authors have assumed that the viral load is undetectable in the patients analyzed under treatment. Considering that this variable is crucial in the course of viral pathogenesis, could they provide the specific data? If this information is unavailable, do they consider that such an assumption should be acknowledged as a limitation of the study that should be addressed in future analyses?
Author Response
Major concerns
- Please include in the Introduction important concepts for readers who are not specialists in the subject. These should include the definition and functional characteristics of TREM2 in its mature and immature forms, expression of the CB2 receptor in microglia (not just in macrophages), abbreviations (carboxy-terminal fragment), and their meanings (HAND, DSM, etc.).
Response:
- We appreciate the reviewer’s suggestion to enhance the Introduction for a broader audience. We have revised the Introduction to include definitions and functional characteristics of TREM2, particularly in its mature, immature, CTF, and soluble forms. Additionally, we have expanded on the role of CB2 receptors in microglia and provided explanations for the implications. We have elaborated on this on:
- Page 2, line 74-85:
TREM2 has multiple isoforms resulting from alternative splicing and proteolytic cleavage, each with distinct molecular weights and functional characteristics. The immature form of TREM2 (~35kDa), a precursor protein primarily localized in the endoplasmic reticulum, is not fully functional but is essential for proper folding, maturation, and transportation to the cell surface[38]. Once membrane-bound, mature TREM2 (~50kDa) can bind to ligands, such as lipids and lipoproteins, to initiate intracellular signaling that leads to microglial survival, proliferation, phagocytosis, and anti-inflammatory responses function[39-41]. Soluble TREM2 (sTREM2) is generated through the proteolytic cleavage of membrane-bound TREM2 by proteases, such as ADAM10 and ADAM17, and also results in the generation of C-terminal fragment (~10kDa)[42]. sTREM2 is capable of binding to ligands that would otherwise interact with membrane-bound TREM2, thus modulating inflammatory responses and microglial activity[43].
- Page 3, line 108-112:
Additionally, activation of the cannabinoid type-2 receptor (CB2) on macrophages can reduce the production of pro-inflammatory cytokines while also promoting microglial motility towards injury sites[62,63]. Targeting cannabinoid receptor 2 (CB2) on periph-eral immune cells may reduce the inflammatory mechanisms implicated in HAND, suggesting a pathway for therapeutic intervention[64].
================================================================================
- Have the authors considered that the model EcoHIV used—according to the evaluation timeline—represents an acute infection in the murine model? Additionally, have they considered that despite EcoHIV invading the brain at low levels and inducing cytokine responses in the brain consistent with mild brain disease in humans, these characteristics might limit the interpretation of the results? This is an experimental animal system that may allow modeling host pathophysiological conditions for mild neurocognitive impairment (NCI) induction in HIV-suppressed people. EcoHIV infection causes NCI but not CD4+ T cell depletion or immunodeficiency in mice. Please also explain how did the authors defined the EcoHIV dose used.
Response:
- The reviewer raises an important point about the limitations of the EcoHIV model in representing chronic HIV infection, particularly in the context of neurocognitive impairment (NCI). We agree that the EcoHIV model, while useful for studying acute viral infection and neuroinflammation, does not fully recapitulate the chronic, immunosuppressive state seen in HIV-infected humans. We acknowledge the limitations of the EcoHIV model, particularly its inability to induce CD4+ T cell depletion and immunodeficiency, which are hallmarks of HIV infection in humans. However, despite these limitations, EcoHIV infection in mice is a valuable experimental system for modeling the pathophysiological conditions associated with mild NCI in HIV-suppressed individuals. We have elaborated on this in the Discussion:
- Page 21, lines 667-672:
The EcoHIV model is well-recognized for its utility in representing the acute phase of HIV infection in mice, characterized by initial viral replication and immune response, including cytokine production[13,80]. While this model simulates an acute infection, it is important to note that EcoHIV does invade the brain, albeit at low levels, leading to cytokine responses that are consistent with mild neurocognitive impairment (NCI) ob-served in humans[81].
- Regarding the dose of EcoHIV used in our study was selected based on previous research identifying the viral load necessary to establish a consistent and measurable infection in the central nervous system (CNS) while allowing for the observation of neuroinflammatory and cognitive outcomes (Cook et al., 2015; Moran et al., 2013). This dosage has been validated as sufficient to induce the desired pathophysiological effects without overwhelming the immune system, thus ensuring the relevance of the findings to the mild NCI observed in human cases.
- Could the authors provide a brief explanation of the rationale for selecting frontal cortex tissues for the analysis of TREM2 expression? Likewise, please specify the sections of the mouse brain that were analyzed for IBA1 and GFAP immunohistochemistry.
Response:
- We appreciate the opportunity to clarify the rationale behind our tissue selection for TREM2 expression analysis and the brain regions used for immunohistochemistry. Frontal cortex tissues were selected for TREM2 expression analysis because this brain region is particularly relevant to neuroinflammatory processes and cognitive functions, which are central to our study. The frontal cortex has been widely studied in the context of neurodegenerative diseases, including Alzheimer's disease, where TREM2 expression has been shown to play a critical role (Jay et al., 2017; Filipello et al., 2018). Additionally, the frontal cortex shares significant pathological and functional connections with the hippocampus (HC), another region implicated in cognitive processes and neuroinflammation (Llorens-Martín et al., 2014). While pure hippocampal lysates for a blot were not available in our study, the use of frontal cortex tissues serves as a robust surrogate due to these established connections.
- Thus, for immunohistochemistry (IHC), we analyzed sections from the hippocampus for IBA1 and GFAP staining. The hippocampus is a key region involved in memory and cognitive functions, and it is particularly susceptible to neuroinflammatory changes. Therefore, it was chosen for IHC to assess the microglial (IBA1) and astrocytic (GFAP) activation, which are critical markers of neuroinflammation (Serrano-Pozo et al., 2013).
================================================================================
- In contrast to so-called homeostatic microglial markers, such as transmembrane protein (TMEM-)119 or P2RY12, IBA1 is also expressed by peripheral macrophages. Even though IBA1 is often characterized as a typical microglial activation marker, it cannot be used to distinguish between functional microglial phenotypes. So, do the authors consider that this marker presents limitations in the evaluation conducted?
Response:
- The reviewer is correct in noting the limitations of IBA1 as a microglial marker, particularly its inability to differentiate between resident microglia and infiltrating macrophages. We acknowledge the limitations of IBA1 as a marker for microglial activation, particularly its expression in peripheral macrophages. Unlike more specific microglial markers such as TMEM119 or P2RY12, which are considered homeostatic and exclusive to microglia, IBA1 is also expressed by peripheral macrophages, which introduces a degree of ambiguity in distinguishing between resident microglia and infiltrating macrophages (Hickman et al., 2018). Additionally, while IBA1 is often characterized as an activation marker for microglia, it does not provide information about the specific functional phenotypes of these cells. Microglia can exhibit a wide range of activation states, ranging from pro-inflammatory to anti-inflammatory, and IBA1 does not differentiate between these functional states (Ito et al., 1998). As such, while IBA1 was useful for identifying the presence of activated microglia, we recognize that it presents limitations in terms of evaluating the specific functional phenotypes of these cells in our experimental context.
- These considerations are included in the discussion, found on page 23 lines 784-787: Moreover, future studies should incorporate additional markers to better distinguish between microglial subtypes and to differentiate microglia from peripheral macro-phages, thereby providing a more nuanced understanding of microglial function and activation in the context of HIV-associated neuroinflammation.
================================================================================
- The interpretation (at the end) of the Results sections 3.3, 3.4, 3.5, 3.6, and 3.7 is not included. The authors' interpretation is valuable for readers prior to its discussion.
Response:
- We agree that including a brief interpretation at the end of each Results section would aid in the reader’s understanding and would provide a logical flow from the presentation of data to the broader discussion of its relevance. We have expanded the interpretation at the end of Results sections 3.3, 3.4, 3.5, 3.6, and 3.7 to provide readers with a clearer understanding of the findings before transitioning to the Discussion. These additions are the following:
- (3.3) Pages 9, lines 391-395: Collectively EcoHIV infection significantly increased astrocyte reactivity in TREM2R47H mice compared to WT mice. Specifically, elevated GFAP signal in the hippocampal sub-regions of EcoHIV-treated TREM2R47H mice indicate heightened astrocyte activation is associated with the TREM2 R47H gene variant following EcoHIV infection.
- (3.4) Pages 9, lines 408-415: These correlation analyses show that TREM2 mRNA levels in MDMs are associated with age and HIV status, with the relationship varying depending on cannabis use. Specifically, these data show a significant positive correlation between TREM2 mRNA and age in PWoH without cannabis use, and this correlation is markedly weaker in PWoH with cannabis use. No such correlation is observed in PWH without cannabis use, however, a significant inverse correlation between TREM2 mRNA levels and age is observed in PWH with cannabis use. Altogether, regulation of TREM2 expression in ex vivo cultured MDMs involves a complex interaction age, cannabis use, and HIV status.
- (3.5) Pages 10, lines 439-445: Across all conditions, these expression analyses show that IL1B treatment consistently reduced TREM2 expression. Co-treatment with CBD and IL1B partially reverses the IL1B-induced reduction in TREM2 expression in both HIV- and HIV+ MDMs, though the response is less robust in HIV+ MDMs. Taken together, differences in TREM2 expression based on cannabis use frequency under these conditions highlight the modulatory effects of CBD on TREM2's response to inflammatory stimuli.
- (3.5) Pages 10, lines 461-472: Overall, we observed key differences in the effects of AM 630 and GW 6471 on TREM2 expression between MDMs from non-cannabis users and cannabis users, with distinct outcomes observed in HIV- and HIV+ groups. In non-cannabis users, C+I cotreatment with AM 630 or GW 6471 revealed significant reductions in TREM2 expression in HIV+ MDMs relative to HIV- MDMs. In cannabis users, however, AM 630 and GW 6471 treatments in the presence of C+I resulted in significantly increased TREM2 expression in HIV+ MDMs relative to HIV- MDMs. These opposing findings further underscore the importance of considering both HIV status and cannabis use when evaluating factors that influence the immunomodulatory role of TREM2 in MDMs. Nevertheless, these results suggest that modulation of IL1B-induced suppression of TREM2 with CBD can be significantly influenced by CB2 and PPARα signaling pathways.
- (3.6) Page 11, lines 520-527: Overall, these findings provide additional implications for understanding how canna-bis use and HIV status affect the gene expression profile of MDMs, particularly in the context of TREM2-related gene expression following CBD treatment. The contrasting expression patterns between HIV+ Naïve and HIV+ Moderate or HIV+ Daily MDMs il-lustrate how cannabis use may suppress some inflammatory pathways while enhancing others. Nevertheless, we show similar gene expression patterns in additional MDMs which substantiates CBD’s impact on TREM2-related gene expression across a broader population.
- (3.7) Pages 12, lines 540-544: All in all, these data highlight the significant impact of CBD on TREM2 dynamics, specifically its ability to counteract the IL1B-induced cleavage of TREM2 and reduce the levels of pro-inflammatory sTREM2. These results demonstrate the potential for CBD to counteract the elevated levels of sTREM2 that contribute to pathological inflammation.
================================================================================
- Check this paragraph (section 3.6): "Relative to HIV+ Naïve, MDMs from HIV+ Moderate PWH displayed an altered transcriptomic profile withXXXand YYY genes up- and down-regulated, respectively".
Response:
- Thank you for pointing this out. The placeholder text "XXX" and "YYY" in Section 3.6 has been replaced with the appropriate gene names, and the paragraph has been revised for clarity.
- This update is on pages 10-11, lines 482-485: Reversed gene expression patterns relative to HIV+ Naïve were seen in HIV+ Moderate MDMs, including upregulation of genes such as STAB1, TMEM176B, CHIT1, SER-PINF1, and TMEM176A while downregulated genes include ITGB3, DPYSL3, CXCL5, GREM1, and CCL15 (Figure 6B).
================================================================================
- Table 1. TBD? please specify. Only one participant has been included in the group HIVneg/moderate.
Response:
- We apologize for the oversight in Table 1. The 'TBD' has been replaced with the finalized information that was not immediately available, such as age/sex/race. Regarding the HIVneg/moderate group, we acknowledge that including only one participant may limit the generalizability of the findings and made a concerted effort to avoid overinterpretation of the data from this group in particular. We have now specified the missing details in Table 1, though demographic data was only available for one participant in the HIVneg/moderate group.
- In Figure 1, the asterisks representing the statistical analysis were not included. In Figure 4, the number of samples analyzed (points in each graph) does not match the details provided in the figure caption.
Response:
- We appreciate the reviewer pointing out these issues. Asterisks for statistical significance in Figure 1 have been included, and we have rechecked the sample numbers in Figure 4 to ensure consistency with the figure caption. These changes, similar to the changes described in the response to the reviewer’s previous comment, were implemented upon receiving the demographic data for donors used in correlation analyses.
================================================================================
- The authors claim to "corroborate" results found in patients with those observed in the murine model using EcoHIV. ("Our observation of decreased total TREM2 expression in response to EcoHIV infection corroborates previous work which shows decreased TREM2 in membrane-enriched fractions of brain homogenates from PWH with HAND" Discussion section). Can this model truly CORROBORATE what is observed in patients?
Response:
- We appreciate the opportunity to clarify our use of the term "corroborate." The intention behind this statement was to highlight the alignment between our findings in the EcoHIV murine model and those observed in patients with HIV-associated neurocognitive disorders (HAND). However, I acknowledge that the term "corroborate" may suggest a stronger level of equivalence between the model and human pathology than is appropriate. The EcoHIV model is indeed a valuable tool for studying HIV-related processes in the brain, but it is important to recognize that it does not fully replicate the complexity of human HAND. Therefore, while our findings in the murine model are consistent with previous observations in human studies, it would be more accurate to state that our results “support” or “align with” these previous findings rather than “corroborate” them.
- This change is on page 20, lines 644-647: This is the first study to show decreased total TREM2 expression in response to EcoHIV infection supports previous work which shows decreased TREM2 in membrane-enriched fractions of brain homogenates from PWH with HAND[50].
================================================================================
- The authors have assumed that the viral load is undetectable in the patients analyzed under treatment. Considering that this variable is crucial in the course of viral pathogenesis, could they provide the specific data? If this information is unavailable, do they consider that such an assumption should be acknowledged as a limitation of the study that should be addressed in future analyses?
Response:
- Thank you for highlighting the absence of viral load data, which could impact the interpretation of our findings. In our study, we assumed that the viral load in the patients analyzed under antiretroviral treatment (ART) was undetectable. This assumption is based on the standard clinical outcome for patients effectively managed on ART, where viral replication is typically suppressed to undetectable levels (Murray et al., 2017). However, we recognize that viral load is a critical variable in the course of HIV pathogenesis, influencing both systemic and central nervous system outcomes. Unfortunately, specific viral load data for the cohort analyzed was not available at the time of the study. We acknowledge that this lack of data represents a limitation of our study, as the precise viral load could impact the interpretation of our findings, particularly concerning the effects of HIV on neuroinflammation and cognitive function. In future analyses, it will be crucial to include and report specific viral load measurements to better contextualize the results and understand their implications in the setting of different levels of viral suppression.
- This addition is on page 23, lines 781-783: We also recognize the critical importance of viral load as a variable in HIV pathogenesis, particularly its impact on neuroinflammation and cognitive function, and future studies should aim to report viral load values when possible.
===============================================================================
Round 2
Reviewer 1 Report
Comments and Suggestions for Authors
The reversion answers the raised issues in my comments.
Reviewer 2 Report
Comments and Suggestions for Authors
The comments made by this reviewer have been adequately addressed, and the modifications introduced in the manuscript are clear, including those that express the inherent limitations of the models and procedures used.